# The Impact of the Injected Mass of the Gastrin-Releasing Peptide Receptor Antagonist on Uptake in Breast Cancer: Lessons from a Phase I Trial of [^99m^Tc]Tc-DB8

**DOI:** 10.3390/pharmaceutics17081000

**Published:** 2025-07-31

**Authors:** Olga Bragina, Vladimir Chernov, Mariia Larkina, Ruslan Varvashenya, Roman Zelchan, Anna Medvedeva, Anastasiya Ivanova, Liubov Tashireva, Theodosia Maina, Berthold A. Nock, Panagiotis Kanellopoulos, Jens Sörensen, Anna Orlova, Vladimir Tolmachev

**Affiliations:** 1Department of Nuclear Therapy and Diagnostic, Cancer Research Institute, Tomsk National Research Medical Center, Russian Academy of Sciences, 634050 Tomsk, Russia; rungis@mail.ru (O.B.); chernov1962@gmail.com (V.C.); r.zelchan@yandex.ru (R.Z.); medvedeva@tnimc.ru (A.M.); 2Research Centrum for Oncotheranostics, Research School of Chemistry and Applied Biomedical Sciences, Tomsk Polytechnic University, 634050 Tomsk, Russiamr.varvashenya@mail.ru (R.V.); 3Department of Pharmaceutical Analysis, Siberian State Medical University, 634050 Tomsk, Russia; 4Department of General Oncology, Cancer Research Institute, Tomsk National Research Medical Center, Russian Academy of Sciences, 634009 Tomsk, Russia; anfioran@yandex.ru; 5Department of General and Molecular Pathology, Cancer Research Institute, Tomsk National Research Medical Center, Russian Academy of Sciences, 634009 Tomsk, Russia; lkleptsova@mail.ru; 6Laboratory of Molecular Therapy of Cancer, Cancer Research Institute, Tomsk National Research Medical Center, Russian Academy of Sciences, 634009 Tomsk, Russia; 7Molecular Radiopharmacy, INRaSTES, NCSR “Demokritos”, 15341 Athens, Greece; maina_thea@hotmail.com (T.M.); nock_berthold.a@hotmail.com (B.A.N.); 8Department of Medicinal Chemistry, Uppsala University, 751 23 Uppsala, Sweden; panagiotis.kanellopoulos@ilk.uu.se (P.K.); anna.orlova@ilk.uu.se (A.O.); 9Radiology and Nuclear Medicine, Department of Surgical Sciences, Uppsala University, 751 85 Uppsala, Sweden; jens.h.sorensen@uu.se; 10Department of Immunology, Genetics and Pathology, Uppsala University, 752 37 Uppsala, Sweden

**Keywords:** radionuclide molecular imaging, gastrin-releasing peptide receptor antagonist, breast cancer, optimal injected mass, SPECT

## Abstract

**Background/Objectives:** Gastrin-releasing peptide receptor (GRPR) is overexpressed in breast cancer and might be used as a theranostics target. The expression of GRPR strongly correlates with estrogen receptor (ER) expression. Visualization of GRPR-expressing breast tumors might help to select the optimal treatment. Developing GRPR-specific probes for SPECT would permit imaging-guided therapy in regions with restricted access to PET facilities. In this first-in-human study, we evaluated the safety, biodistribution, and dosimetry of the [^99m^Tc]Tc-DB8 GRPR-antagonistic peptide. We also addressed the important issue of finding the optimal injected peptide mass. **Methods:** Fifteen female patients with ER-positive primary breast cancer were enrolled and divided into three cohorts receiving [^99m^Tc]Tc-DB8 (corresponding to three distinct doses of 40, 80, or 120 µg DB8) comprising five patients each. Additionally, four patients with ER-negative primary tumors were injected with 80 µg [^99m^Tc]Tc-DB8. The injected activity was 360 ± 70 MBq. Planar scintigraphy was performed after 2, 4, 6, and 24 h, and SPECT/CT scans followed planar imaging 2, 4, and 6 h after injection. **Results:** No adverse events were associated with [^99m^Tc]Tc-DB8 injections. The effective dose was 0.009–0.014 mSv/MBq. Primary tumors and all known lymph node metastases were visualized irrespective of injected peptide mass. The highest uptake in the ER-positive tumors was 2 h after injection of [^99m^Tc]Tc-DB8 at a 80 µg DB8 dose (SUV_max_ 5.3 ± 1.2). Injection of [^99m^Tc]Tc-DB8 with 80 µg DB8 provided significantly (*p* < 0.01) higher uptake in primary ER-positive breast cancer lesions than injection with 40 µg DB8 (SUV_max_ 2.0 ± 0.3) or 120 µg (SUV_max_ 3.2 ± 1.4). Tumor-to-contralateral breast ratio after injection of 80 μg was also significantly (*p* < 0.01, ANOVA test) higher than ratios after injection of other peptide masses. The uptake in ER-negative lesions was significantly lower (SUV_max_ 2.0 ± 0.3) than in ER-positive tumors. **Conclusions:** Imaging using [^99m^Tc]Tc-DB8 is safe, tolerable, and associated with low absorbed doses. The tumor uptake is dependent on the injected peptide mass. The injection of an optimal mass (80 µg) provides the highest uptake in ER-positive tumors. At optimal dosing, the uptake was significantly higher in ER-positive than in ER-negative lesions.

## 1. Introduction

Gastrin-releasing peptide receptor (GRPR) is a G-protein-coupled receptor involved in regulating hormone release, body temperature, and contraction of smooth musculature, as well as in cellular proliferation and differentiation [1]. Ectopic overexpression GRPR is documented in several malignancies including prostate, breast, gastrointestinal, colon, and non-small cell lung carcinomas [2]. Bombesin is an amphibian counterpart of the human gastrin-releasing peptide, the natural mammalian ligand of GRPR, binding with high affinity to human GRPR and neuromedin B receptor [3]. Radiolabeled bombesin analogs have been proposed and evaluated for imaging and therapy of GRPR-expressing cancers [3,4]. An interest in radionuclide targeting of GRPR in breast cancer increased during recent years [5], owing to several factors. First, GRPR is overexpressed, often at a high level, in a large fraction of malignant breast tumors [6,7], and its expression is typically preserved in metastases [6,7,8,9,10]. Furthermore, developing high-affinity bombesin analogs with a GRPR antagonist profile enabled addressing of biosafety concerns associated with adverse acute physiological effects elicited after injection of GRPR agonists. In addition, given that GRPR antagonists turned out to be more resistant to degrading proteases compared with their agonist counterparts, the shift toward antagonist-based radioligands led to the development of efficient agents, primarily for the visualization and treatment of prostate cancer [11,12]. Apparently, this experience is also essential for the implementation of such agents for theranostics of breast carcinomas. One of the challenges in the therapeutic use of GRPR-targeting agents is the heterogeneity of the target’s expression [9], which might result in false-negative biopsy findings because of sampling errors. Although in vivo radionuclide molecular imaging does not provide such spatial resolution as with ex vivo immunohistochemistry (IHC) analysis of biopsy material, it can potentially provide information concerning the whole tumor and its metastases. Thus, molecular imaging might overcome GRPR heterogeneity issues and enable the selection of patients with tumors having sufficiently high GRPR expression for radionuclide therapy. In addition, radionuclide molecular imaging is non-invasive and permits repetitive determination of expression, addressing the issue of temporal heterogeneity.

Interestingly, GRPR might act also as a reporter for elevated levels of estrogen receptor (ER) expression in luminal A and luminal B molecular subtypes of invasive ductal breast carcinomas. Overexpression of ER is a feature of these cancer subtypes and this overexpression is a predictive biomarker for response to hormonal therapy [13]. The major issue for prediction is a conversion of ER status in distant metastases [14], which might result in over- and undertreatment of patients with metastatic or recurrent disease. A possible solution might be PET/CT imaging of ER expression using 16α-[^18^F]fluoro-17β-estradiol (FES), which demonstrated a good agreement between imaging and immunohistochemistry results in several retrospective and prospective studies [15,16,17]. However, this tracer has a high uptake in the liver due to its lipophilicity and metabolism, which complicates detecting ER-positive hepatic metastases [16]. This is a serious limitation taking into account the high frequency of liver metastases of breast cancer [18]. At the same time, there is an exceptionally strong correlation between the expression of GRPR and ER both on mRNA and protein levels [7,8,10,19]. A small series of clinical PET studies performed by researchers from Freiburg using the GRPR antagonist radioligand [^68^Ga]Ga-RM2 confirmed the validity of the use of bombesin analogs for the detection of ER [20,21,22,23]. An independent meta-analysis of data from this series [24] showed the lesion detectability of 93% for ER-positive breast cancer. In addition, the standardized uptake values (SUV_max_) for histologically verified ER-negative tumors were significantly lower than for ER-positive ones.

Thus, an evaluation of GRPR expression level in breast cancer might provide essential predictive information for the selection of, e.g., GRPR-targeted radionuclide therapy or, possibly, therapy based on blocking ER signaling.

The mainstream in the development of novel tracers for the visualization of GRPR is based mainly on positron-emitting nuclides. This approach utilizes superior resolution and straightforward in vivo quantification of radioactivity concentration typical for positron emission tomography (PET). However, PET imaging facilities tend to be more concentrated in high-income countries due to the high costs associated with the equipment, maintenance, and the need for nearby cyclotron installation to produce the necessary radiotracers. This often makes PET imaging less accessible in low- and middle-income countries where economic and logistical challenges are more pronounced [25]. SPECT/CT cameras are generally more accessible and widespread [25]. SPECT imaging often uses the inexpensive generator-produced radionuclide Tc-99m, which makes it more practical for regions with limited resources. The disparities in the availability of these imaging technologies highlight the need for the development of [^99m^Tc]Tc-labeled imaging probes to improve access to advanced diagnostic tools in underserved areas. Importantly, modern SPECT/CT scanners have significantly improved over time. With the built-in capability to correct for scatter, attenuation, and partial volume effects, they achieve an in vivo quantification accuracy of around 5%. This motivated the development and preclinical evaluation of ^99m^Tc-labeled agents for GRPR imaging in several research centers [26,27,28,29,30,31]. We have earlier reported the preclinical [30] and clinical [32] evaluation of the GRPR antagonist-based tracer [^99m^Tc]Tc-maSSS-PEG2-RM26 aimed, first and foremost, at the staging of prostate cancer. To minimize the interfering activity from the urinary bladder, that tracer was designed to be partially excreted via bile. During the Phase I evaluation, [^99m^Tc]Tc-maSSS-PEG2-RM26 enabled visualization of both primary prostate and breast cancer tumors as well as lymph node metastases [32]. However, the hepatobiliary excretion resulted in elevated liver uptake, making [^99m^Tc]Tc-maSSS-PEG2-RM26 less suitable in the case of breast cancer with frequent liver metastases. Another GRPR antagonist-based imaging agent, designated [^99m^Tc]Tc-DB8 (Figure 1), demonstrated a much lower level of hepatobiliary excretion in preclinical studies [28]. Therefore, we selected [^99m^Tc]Tc-DB8 for clinical evaluation as a potential radiopharmaceutical for SPECT visualization of GRPR in breast cancer. The first step in this evaluation is an assessment of safety, distribution, and dosimetry.

There was another important consideration during the planning of this study. The data from our clinical studies concerning the visualization of HER2 using different radiolabeled scaffold proteins [33,34,35,36] emphasize the importance of the selection of the optimal mass of the injected probe to obtain the highest accumulation of activity in lesions, the best imaging contrast, and clear discrimination of HER2-positive and HER2-negative tumors. Analysis of the literature suggested that this factor has not been taken into consideration in clinical evaluations of radiolabeled GRPR antagonists. Of course, the injected activity has been always reported, but the injected mass has not been disclosed explicitly (e.g., [21,22,23,37,38,39,40,41], although the descriptions of radiochemistry suggest that the injected mass was below (sometimes well below) 50 µg. Indeed, when such information was available, the injected mass was below 50 µg [20,42,43,44]. Although the imaging data were promising, there was a risk that the tumor uptake and tumor-to-organ ratios did not reach the maximum possible values, which might affect both the sensitivity and specificity of imaging. Thus, our intention in this study was also to evaluate the impact of injected mass on the uptake of GRPR antagonists in tumors and in normal tissues, as well as the impact on dosimetry.

According to the trial registration (ClinicalTrials.gov Identifier: NCT05940298), “The primary objectives were: to assess the distribution of [^99m^Tc]Tc-DB8 in normal tissues and tumors at different time intervals; to evaluate dosimetry of [^99m^Tc]Tc-DB8, to study the safety and tolerability of the drug [^99m^Tc]Tc-DB8 after a single injection in a diagnostic dosage. The secondary objective was to compare the obtained [^99m^Tc]Tc-DB8 SPECT imaging results with the data of CT and/or MRI and/or ultrasound examination and immunohistochemical (IHC) studies in breast cancer patients.”

## 2. Materials and Methods

### 2.1. Patients

This was a prospective, open-label, non-randomized Phase I diagnostic study in patients with untreated primary breast cancer (ClinicalTrials.gov Identifier: NCT05940298). The study protocol was approved by the Scientific Council of Cancer Research Institute and Board of Medical Ethics and the Tomsk National Research Medical Center of the Russian Academy of Sciences (protocol №13 from 2 August 2022), and all patients signed a written informed consent to participate in the study and use their images in publications.

Inclusion criteria were as follows: age 18 to 75 years; clinical and radiological diagnosis of breast cancer with histological verification; hematological, liver and renal function tests results within normal limits; a negative pregnancy test for all patients with childbearing potential; patient capability to undergo the diagnostic investigations to be performed in the study; informed consent.

Exclusion criteria were as follows: active current autoimmune disease or history of autoimmune disease; active infection or history of severe infection within the previous 3 months (if clinically relevant at screening); known HIV-positive or chronically active hepatitis B or C; administration of another investigational medicinal product within 30 days of screening; ongoing toxicity > grade 2 from previous standard or investigational therapies, according to the US National Cancer Institute’s guidelines.

Morphological verification of primary tumors was performed in all patients. The material for histology was collected by core biopsies from the upper, middle, and lower third of the primary tumor. Lymph node (LN) metastases were confirmed by cytology using fine-needle biopsy in all patients.

For the evaluation of estrogen receptor (ER) and progesterone receptor (PR) status, immunohistochemical analysis was performed on formalin-fixed, paraffin-embedded tissue sections using the Leica Bond Max automated staining system, following the standard protocol. Monoclonal antibodies against ER (clone 1D5) and PR (clone PgR 636) were utilized (Dako, Carpinteria, CA, USA). The histological sample slides were digitized using Aperio AT2 (Leica Biosystems, Wetzlar, Germany). QuPath v0.6.0 software was used to analyze slides. ER and PR expression were assessed using the Allred score, a semi-quantitative method that combines the proportion of positive cells and the intensity of staining. The proportion score (ranging from 0 to 5) is determined by the percentage of positive tumor cells, while the intensity score (ranging from 0 to 3) reflects the staining intensity. The total Allred score ranges from 0 to 8, with scores ≥3 considered positive. The proportion score is assigned as follows: 0 (0%), 1 (<1%), 2 (1–10%), 3 (11–33%), 4 (34–66%), and 5 (67–100%). The intensity score is assigned as 0 (no staining), 1 (weak), 2 (moderate), or 3 (strong).

For the evaluation of GRPR expression, staining with a GRPR monoclonal antibody (1:500 dilution, clone 18H31L38, Thermo Fisher Scientific, Waltham, MA, USA, Cat. No. 703928) was performed using the Leica Bond Max automated staining system, with appropriate positive and negative controls. The stained samples were evaluated at ×40 magnification using Aperio AT2 (Leica Biosystems, Wetzlar, Germany). To assess the co-expression of estrogen receptor and GRPR, TSA-modified immunohistochemistry was performed using the Opal 3-plex kit (Akoya, Marlborough, MA, USA), following the manufacturer’s protocol. The multiplex sample slides were digitized using Vectra 3 (Akoya, Marlborough, MA, USA). InForm 3.0 software was used to analyze slides. Results were scored to categorize cells into three distinct populations: ER-GRPR+, ER+GRPR-, and ER+GRPR+. These scores were calculated as a proportion of all tumor cells present in the analyzed tissue sections.

The level of expression of HER2 in biopsy samples was determined by immunohistochemistry using Herceptest (DAKO). In the case of a score of 2+, fluorescent in situ hybridization (FISH) with LSI HER2/neu (17q12)/CEP17 probe (Leica) was used to assess HER2 amplification. The tumors were classified as HER2-positive in cases of IHC score 3+ or IHC score 2+ and are FISH-positive, and HER2-low or -negative if below these scores.

Nineteen female patients were enrolled (Table 1; Figure 2). Patients with ER-positive primary tumors were divided into three cohorts injected with [^99m^Tc]Tc-DB8 and corresponding each to a mass of 40 μg, 80 μg, or 120 μg of DB8. Recruitment of patients into a cohort with a higher injected dose was initiated no earlier than the accomplishment of the safety evaluation of the preceding cohort with a lower injected dose. In each cohort, patients were enrolled consecutively. Additionally, four patients with ER-negative primary tumors were enrolled. These patients were injected with 80 μg of ^99m^Tc-labeled DB8. There was no significant difference between body weight of patient’s cohorts (Appendix A).

For all patients, mammography (Pristina Serena Operator Manual Extract, GE HealthCare Technologies Inc., Chicago, IL, USA), bone scan (Siemens Symbia Intevo Bold SPECT/CT hybrid scanner, Siemens Healthineers, Erlangen, Germany) using [^99m^Tc]Tc-pyrophosphate, chest CT (Siemens Somatom Confidence, Siemens Healthineers, Erlangen, Germany), and ultrasound imaging of the breast, regional lymph nodes, and liver (GE LOGIQ E9, GE HealthCare Technologies Inc., Chicago, IL, USA) were performed. The size of the primary tumor and metastatic lymph nodes were measured using ultrasound. Physical examination, clinical lab analysis, and electrocardiogram were additionally performed immediately before imaging and 2 days later, after the last scan.

### 2.2. Radiopharmaceutical and Imaging Protocol

DB8 for this study was produced by the contract manufacturing organization piCHEM Forschungs und Entwicklungs GmbH, Raaba-Grambach, Austria. The substance was delivered and stored as a freeze-dried powder. The analysis (HPLC and MALDI-TOF mass-spectroscopy) demonstrated that the purity of DB8 was more than 95%. Labeling of DB8 was performed under aseptic conditions in the following steps: The [^99m^Tc]TcO_4_^−^ containing generator eluate (600–800 MBq, 300–500 µL) was mixed with 25 µL 0.5 M phosphate buffer, pH 12.0, and 5 µL 0.1 M trisodium citate. To this solution, 30, 60, or 90 nmol of DB8 (as a solution in water) was added followed by 40 µg SnCl_2_ × 2H_2_0 dissolved in 20 µL ethanol. After 30 min incubation at room temperature, 5 mL 1 M NaH_2_PO_4_ was added to adjust the pH to the range between 7.0 and 9.0, and 100 µL ethanol was added. For injection, a required activity was formulated with unlabeled DB8 to provide an injected peptide mass dose of 40, 80, or 120 µg. The sterile filtration was performed using a 0.2 µm filter (Sterifix, B.Braun, Melsungen, Germany).

The average radiochemical yield was 97 ± 2%. The injected activity was 334 ± 84, 415 ± 102, and 370 ± 72 MBq for the groups injected with peptide mass doses of 40, 80, and 120 µg, respectively. Monitoring of safety, thorough assessment of vital signs, ECG, and physical examination were performed during imaging visits (0–24 h after injection) and 3–7 days after injection.

Imaging was performed using a Siemens Symbia Intevo Bold scanner equipped with a high-resolution, low-energy collimator. Anterior and posterior planar whole-body imaging (at a scan speed of 12 cm/min, 1024 × 256 pixel matrix) was performed at 2, 4, 6, and 24 h after injection of [^99m^Tc]Tc-DB8. SPECT/CT scans (SPECT: 60 projections, 20 s each, stored in 256 × 256 pixel matrix/CT: 130 kV, effective 36 mAs) covering the area from the neck to liver were performed 2, 4, and 6 h after injection of [^99m^Tc]Tc-DB8. SPECT images were reconstructed using xSPECT (Siemens) protocol based on the ordered subset conjugate gradient (OSCG) method (24 iterations, 2 subsets) with a 3D Gaussian FWHM 10 mm filter (Soft Tissue). The images were processed using the proprietary software package Syngo.via Workstation (Siemens Healthineers).

Dosimetry was evaluated as described in [44]. Briefly, regions of interest (ROI) were drawn over organs of interest and over the whole body on the anterior and posterior whole-body images of patients injected with [^99m^Tc]Tc-DB8; a geometric mean of counts at 2, 4, 6, and 24 h was calculated for each ROI. To assess elimination kinetics in blood, a ROI was placed over the heart. For quantification, a known activity of Tc-99m in a water-filled phantom in combination with Chang’s correction was used. Data were fitted by single exponential functions and residence time was calculated as areas under fitted curves using GraphPad Prism 8.0 (GraphPad Software, San Diego, CA, USA). Absorbed doses were calculated by OLINDA/EXM 1.1 using Adult Female phantom.

Maximal standard uptake values (SUV_max_) were calculated in primary tumors, lymph nodes, liver, as well as in contralateral symmetric regions of the breast and contralateral symmetric regions for lymph node metastases for determination of tumor-to-contralateral breast and lymph node metastases to background ratios at 2, 4, and 6 h after injection.

### 2.3. Statistics

Values are reported as mean ± SD. Differences between uptakes in organs at different injected mass of the DB8 were analyzed using the nonparametric Mann–Whitney U test or one-way Analysis of Variance (ANOVA), when more than three groups were compared. A 2-sided *p* value of less than 0.05 was considered significant. Paired *t*-tests were used to analyze differences between uptakes in organs at different time points. The statistical analysis was performed using Prism 9 for Windows (GraphPad Software, San Diego, CA, USA).

## 3. Results

All injections were well tolerated. No clinically significant changes in vital signs or results of blood and urine analyses before and after [^99m^Tc]Tc-DB8 injection were observed.

Information concerning the distribution of [^99m^Tc]Tc-DB8 is presented in Figure 3 and Figure 4 as well as Table 2. By 2 h after injection, no major blood vessels were visualized, i.e., activity was distributed from blood to organs and tissues and partially cleared from the body. The half-lives of further elimination were 2.5 ± 0.7 h, 3.0 ± 1.1 h, and 3.3 ± 0.4 h for injected masses 40 µg, 80 µg, and 120 µg, respectively (no significant difference, *p* > 0.05, ANOVA with Tukey correction for multiple comparisons). The tracer was eliminated predominantly via the kidneys, and high activity was observed in the urinary bladder. In patients 10 and 7, gallbladders were clearly visualized. There was also some activity accumulation in the content of the intestines, which indicated some degree of hepatobiliary excretion. The tissues and organs with the highest accumulation were the mammary glands, kidney, liver, lung, pancreas, and small intestines. There was no significant difference in the activity uptake of [^99m^Tc]Tc-DB8 in these organs between cohorts injected with 40 µg and 80 µg DB-8 (*p* > 0.05). The uptake significantly decreased between time points (*p* < 0.05, paired *t*-test). The pancreatic uptake was significantly lower at 2 and 4 h after injection of 120 µg than after injection of 40 µg (Figure 5).

The data concerning dosimetry in breast cancer patients are presented in Table 3. There was no significant difference (*p* > 0.05) between absorbed doses after injection of 40 µg and 80 µg [^99m^Tc]Tc-DB8/DB8. The organs with the highest absorbed doses were the breast, kidneys, pancreas, thyroid, and urinary bladder. The absorbed dose to the pancreas was significantly lower after injection of 120 µg than after injection of 40 µg. The effective dose was higher after injection of 120 µg than after injection of 40 µg.

All primary lesions were visualized by SPECT/CT at all time points during the day of injection (2, 4, and 6 h after injection) of [^99m^Tc]Tc-DB8 (Figure 3 and Appendix A). Injection of [^99m^Tc]Tc-DB8 with 80 µg DB8 provided significantly (*p* < 0.01, ANOVA test) higher uptake in primary ER-positive breast cancer lesions than injection with 120 µg DB8 at 2 h p.i. as well as injection with 40 µg DB8 at all time points (Figure 6 and Figure 7). Tumor-to-contralateral breast ratios 2 h after injection of 80 μg were significantly (*p* < 0.01, ANOVA test) higher than ratios after injection of other peptide masses (Figure 8). Injection of 80 µg DB8 provided a significantly (*p* < 0.01, ANOVA test) higher tumor-to-liver ratio at 4 and 6 h p.i. (Figure 9). Overall, injection of [^99m^Tc]Tc-DB8 with a DB8 mass of 80 µg appeared to be the best choice for imaging.

To select an optimal time point for imaging, both tumor uptake and imaging contrast were compared at different time points after injection of [^99m^Tc]Tc-DB8 with 80 μg DB8. There was no significant difference (*p* = 0.3, paired *t*-test) between tumor uptake at 2 h (SUV_max_ = 5.3 ± 1.4) and 4 h (SUV_max_ = 4.8 ± 1.9) after injection of [^99m^Tc]Tc-DB8 with 80 µg DB8, but the uptake at 6 h after injection (SUV_max_ = 4.0 ± 1.6) was significantly lower (Appendix A). There was no significant difference (*p* > 0.05, paired *t*-test) between tumor-to-contralateral breast and tumor-to-liver ratios at 2, 4 and 6 h after injection of [^99m^Tc]Tc-DB8 with 80 µg DB8.

Additionally, four patients with triple-negative breast cancer were included in the study to investigate the feasibility of using a tracer for discrimination between ER-positive and ER-negative lesions. SPECT/CT was performed by administering [^99m^Tc]Tc-DB8 with 80 μg DB8 at 2 h p.i. for all patients (Figure 10 and Figure 11). The uptake in ER-positive primary lesions (SUV_max_ = 5.3 ± 1.2) was significantly (*p* = 0.016, Mann–Whitney test) higher compared to the uptake in ER-negative tumors (SUV_max_ = 2.0 ± 0.8) (Figure 10A). Preliminary data from this study show that the use of a SUV_max_ cut-off value of 3.12 would enable discrimination between ER-positive and ER-negative primary tumors.

All known node metastases were visualized in all patients, regardless of the estrogen receptor status and DB8 dose (Figure 12). Moreover, there were obvious signs of metastasis in the axillary region in two patients injected with [^99m^Tc]Tc-DB8 with a corresponding DB8 dose of 80 μg 2 h after injection (SUV_max_ 1.74, metastasis-to-contralateral site ratio 4.9; and SUV_max_ 4.58, metastasis-to-contralateral site ratio 19.9) (Figure 13). The cytological verification of the lymph nodes involvement was negative (cN0) at the imaging time for these patients. However, the results of the morphological examination after surgery showed that there were metastases in axillary lymph nodes (pN1) in both cases.

Bone scintigraphy of Patient 19 did not reveal any abnormal accumulation of activity (Figure 14A). After injection of [^99m^Tc]Tc-DB8 with 120 μg DB8, an accumulation was detected in the projection of the fifth rib on the right (Figure 14B). A site of lytic destruction (19 × 5 mm) suspicious of metastasis was visualized in the projection of the fifth rib when reviewing the chest CT (Figure 14C). The suspicious area was placed under dynamic control with no morphological verification. After the combined treatment, a site of sclerotic changes (26 × 5 mm) was visualized by chest CT in the projection of the fifth rib (Figure 14B), which corresponded to the therapeutic response of metastasis to the treatment.

### Immunohistochemistry Evaluation

The antibody used in our study enabled the detection of both membranous and cytoplasmic expression of the GRPR protein. ER-negative malignant cells demonstrated no membranous staining, although some cytoplasmic staining was observed. Both membranous and cytoplasmic expression of GRPR was detected in ER-positive cancer cells. Some membranous staining of immune cells infiltrating both ER-positive and ER-negative tumors were also detected (Figure 15).

Multiplex analysis of ER and membranous GRPR co-expression in ER-positive tumors was performed on a subset of samples (Figure 16). The results demonstrated that cells with the ER-/GRPR+ phenotype were not observed in tumor tissue. ER+/GRPR- cells were predominated, followed by ER+/GRPR+ cells, while ER-/GRPR- cells were present in minimal numbers.

## 4. Discussion

Both pan-carcinoma overexpression of GRPR and the demonstration of its radionuclide visualization feasibility [45] prompted active research and development of targeting agents for radionuclide diagnostics and therapy of GRPR-expressing tumors. However, the adverse effects caused by the agonist-based GRPR radioligands [46] reduced the enthusiasm of the scientific community. The shift toward GRPR antagonists changed the situation. Clinical investigations demonstrated that the PET tracers based on antagonists are well tolerated [12,20,38,39,41,42]. Of note, a direct intra-patient comparison showed that the antagonist-based [^68^Ga]Ga-RM26 displayed higher tumor uptake and was capable of detecting more primary lesions and lymph node metastases than its agonist counterpart, [^68^Ga]Ga-BBN [39]. These data indicate that the imaging probes for positron emission tomography would be successfully translated into clinics. Our study was initiated to ensure that the radionuclide imaging of GRPR-expressing tumors would also be available outside high-income regions. The aim was to assess the performance in both breast and prostate cancers. Taking into account that the previous study showed a noticeable difference in the biodistribution and dosimetry of [^99m^Tc]Tc-maSSS-RM26 in male and female patients [32], most likely due to gender-related differences in body size and physiology, the data are presented separately. In the case of breast cancer, this Phase I study was performed in female patients with primary tumors, with potential lymph node involvement. Biopsy material is taken routinely from these lesions, with the advantage of enabling the determination of their receptor status without the need for additional biopsies. A number of clinical studies showed the absence of adverse effects after injection of GRPR-antagonistic probes. The impact of the mass of injected peptide could be evaluated for the first time in the present Phase I study, because it might affect both the biodistribution and the absorbed doses of the tracer to normal tissues.

This study confirmed that the [^99m^Tc]Tc-DB8 injections were safe and well tolerated at all injected peptide masses. The injected activities were selected to enable good counting statistics, permitting accurate dosimetry calculations up to 24 h after injection. Still, the absorbed equivalent dose (4.6 mSv) after injection of the optimal DB8 (80 µg, 60 nmol) mass was comparable with the absorbed dose after injection of [^18^F]FDG for visualization of breast cancer, 2.5–4.3 mSv [47,48]. Considering that the clear SPECT visualization was already observed 2 h after injection, the injected activity might be appreciably reduced further leading to minimizing the population dose from breast cancer diagnostics using [^99m^Tc]Tc-DB8.

[^99m^Tc]Tc-DB8 had a clearly more favorable excretion pattern compared with [^99m^Tc]Tc-maSSS-RM26 for imaging of potential visceral metastases of breast cancer. For example, the hepatic uptake of [^99m^Tc]Tc-DB8 was twofold lower at 2 h after injection, and the uptake in the small intestines was fivefold lower. Another essential finding was a significant reduction in pancreatic uptake after injection of 120 µg than after injection of 40 µg (Figure 5). It is well known that pancreatic cells express GRPR. Thus, the reduction suggests saturable (i.e., GRPR-specific) [^99m^Tc]Tc-DB8 uptake in vivo.

An important finding of this study was the strong effect of the injected DB8 mass on the uptake in tumors (Figure 6 and Figure 7) and tumor-to-normal organ ratios (Figure 8 and Figure 9). The tumor uptake (2 h post-injection) was significantly higher after injection of 80 µg DB8. The increase in SUV_max_ was more than twofold compared to the value after injection of 40 µg. The twofold difference in tumor-to-contralateral breast ratio was also observed in this clinical study. The further increase in the injection mass to 120 µg did not improve the tumor uptake or tumor-to-organ ratios, presumably due to the partial saturation of binding sites in tumors by unlabeled DB8. This finding is consistent with preclinical data obtained in mice for the GRPR agonist [^67^Ga]Ga-DOTA-BZH3 [49] and the GRPR antagonist [^111^In]In-NOTA-P2-RM26 [50]. In both cases, injection of 15 pmol per mouse resulted in significantly higher tumor uptake compared with injections of smaller or larger masses. However, the effect was not as strong as in this clinical study; the increase was between 20 and 56% for the antagonist and about 40% for the agonist. Thus, the best visualization in this study was enabled by injection of the peptide mass, which was higher than the mass typically used in clinical studies evaluating imaging using GRPR antagonists. In principle, this agreed with the data obtained for different classes of peptide-based radionuclide imaging probes. Kooij and co-authors [51] published in 1994 an abstract reporting that injection of 1 µg of [^111^In]In-Octreoscan (somatostatin receptor agonist) resulted in poor detection of neuroendocrine tumors, and injection of [^111^In]In-Octreoscan containing 5–200 µg unlabeled DTPA-octreotide provided better visualization. The full paper was not published, but this information was repeated in papers reporting preclinical investigations of this issue [52,53]. The data from rat models confirmed that both tumor uptake and tumor-to-organ ratios were the highest, though not in the case of the lowest injected mass. A reduction in tracer uptake in normal somatostatin receptor-expressing organs, leading to a decrease in background and making it more available for tumors, was proposed as an explanation for this phenomenon. Similar phenomena were observed in clinics for the [^68^Ga]Ga-labeled somatostatin receptor agonist DOTATOC [54], but not for the antagonist OPS202 [55]. This points out that the observations valid for one type of tracer might not always be directly translated to another, and a careful evaluation is required.

Strictly speaking, the injected mass per kg of body weight would be more appropriate in the optimization. Selection of the injection mass in this case might help to compensate for unusually high or unusually low body weight (see Appendix A). However, we consider that the best approach to clinical translation would be the development of a freeze-dried kit with a fixed mass of the tracer. This would assist in avoiding mistakes during formulation. Thus, our approach is a compromise between the best possible dosing and the eventual availability of a [^99m^Tc]Tc-DB8 established formulation in hospitals, irrespective of personnel training level.

The pioneering study of GRPR imaging using [^68^Ga]Ga-RM2/PET has shown that the uptake (SUV_max_) of that tracer was significantly higher in ER-positive lesions than in ER-negative ones [20]. The authors proposed that [^68^Ga]Ga-RM2/PET might be used as an alternative to [^18^F]F-FES-PET/CT for tumor staging and/or for treatment monitoring. Much lower liver and intestinal [^68^Ga]Ga-RM2/PET uptake was pointed out as a noticeable advantage. This study suggests that [^99m^Tc]Tc-DB8 might be also used for this purpose (Figure 10 and Figure 11). Moreover, a need for such a tracer is urgent, as there is no alternative to SPECT/CT visualization of ER-expressing visceral metastases. The exact mechanism of [^99m^Tc]Tc-DB8 uptake in tumors is intriguing. The immunohistochemistry data show that only ER-positive breast cells have an expression of GRPR on their membranes, where it is accessible for targeting peptides. However, there is also some degree of GRPR expression on membranes of tumor-infiltrating immune cells, which might explain the uptake of [^99m^Tc]Tc-DB8 in ER-negative tumors.

An obvious limitation of this study is the small patient population typical for Phase I trials. Thus, the statistical power is insufficient for strong statements and predictions. Still, we can envision several clinical applications where SPECT imaging using [^99m^Tc]Tc-DB8 might be helpful after further clinical development. These applications include assessing GRPR expression to identify patients who are more likely to respond to GRPR-targeted radionuclide therapies; non-invasive assessment of ER expression in metastatic lesions; therapy response monitoring; imaging of invasive lobular carcinoma metastases, which often have low uptake of [^18^F]-FDG but high expression of ER.

In conclusion, this study shows that [^99m^Tc]Tc-DB8 is not associated with adverse effects after injection with up to 120 µg DB8. Equivalent doses are comparable with doses from the use of [^18^F]-FDG but might be reduced further if successful imaging is confirmed at 2 h or earlier in further studies. The dosimetry after injection of 120 µg DB8 was definitely less favorable in terms of effective dose to a patient, although it was still safe. The tumor uptake and tumor-to-organ ratios strongly depend on the injected mass of cold DB8, and the mass of 80 µg appears to be optimal. The results of this study clearly suggest that the injected mass of any radiolabeled BBN-derived GRPR antagonist has to be optimized to avoid underperformance.

## Figures and Tables

**Figure 1 pharmaceutics-17-01000-f001:**
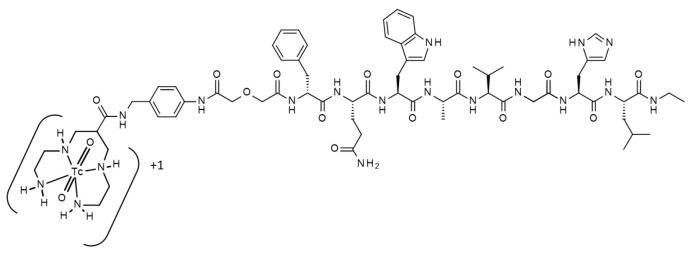
Structure of [^99m^Tc]Tc-DB8.

**Figure 2 pharmaceutics-17-01000-f002:**
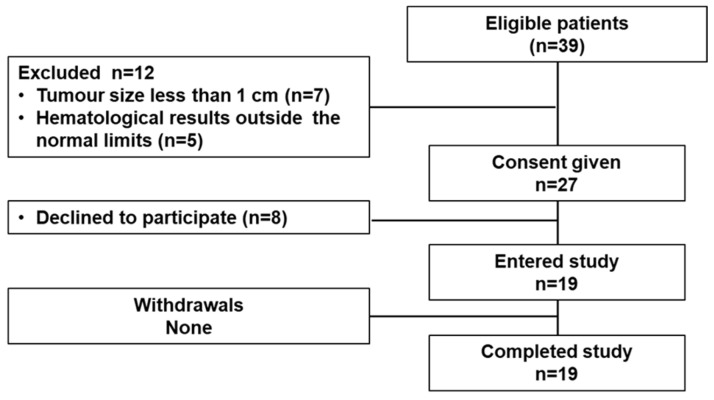
STARD diagram.

**Figure 3 pharmaceutics-17-01000-f003:**
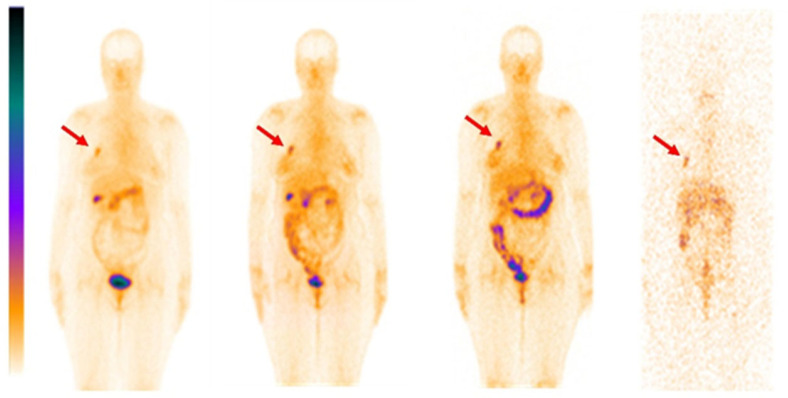
Representative distribution of [^99m^Tc]Tc-DB8 with 80 µg DB8 (Patient 7, anterior projections, planar imaging) at 2, 4, 6, and 24 h. The red arrow points at the tumor. A linear relative scale (normalized at the maximum activity in the image) is applied.

**Figure 4 pharmaceutics-17-01000-f004:**
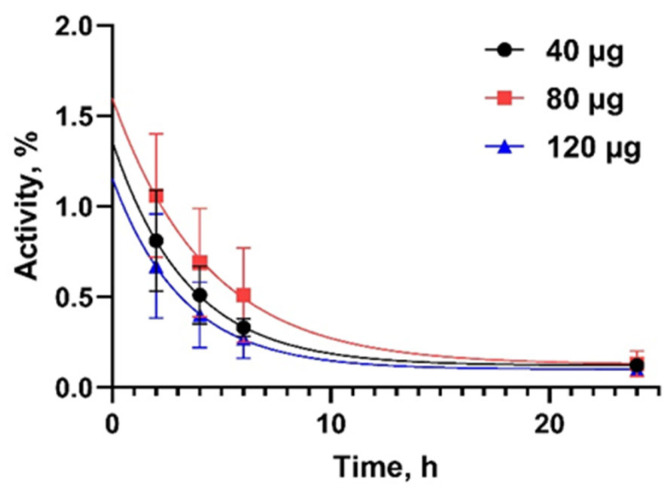
Kinetics of [^99m^Tc]Tc-DB8 elimination from blood associated with DB8 masses of 40, 80, and 120 µg.

**Figure 5 pharmaceutics-17-01000-f005:**
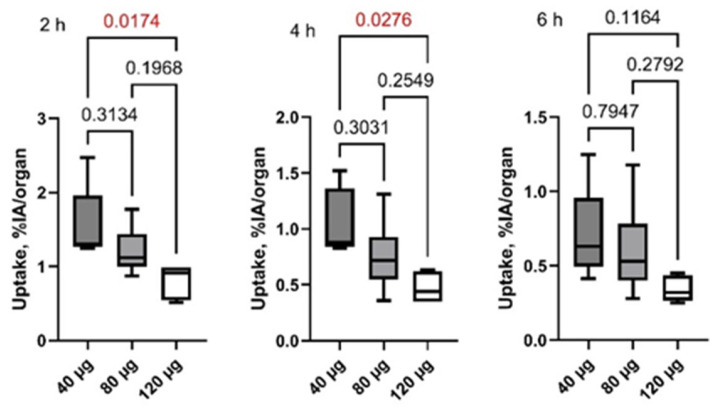
Pancreatic uptake (%IA/organ) after injection of [^99m^Tc]Tc-DB8 with DB8 masses of 40, 80, and 120 µg. Statistical significance was evaluated using one-way ANOVA and is indicated in red.

**Figure 6 pharmaceutics-17-01000-f006:**
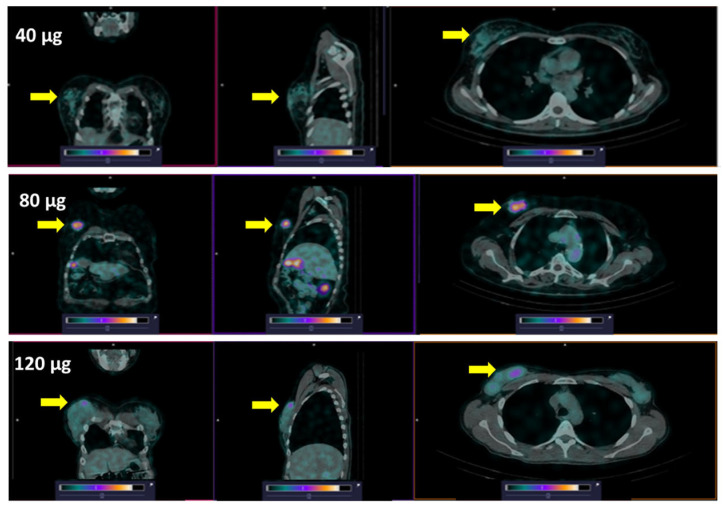
Representative SPECT/CT images (Patients 2, 6, and 15) of primary ER-positive tumors 2 h after injection of [^99m^Tc]Tc-DB8 with 40, 80, and 120 μg DB8. Arrows point at tumors. The upper setting of a linear intensity scale is adjusted to SUV 3 in all images. Note: The upper setting in this Figure was selected to show clearly the superiority of the tumor uptake after injection of 80 μg DB8. Appendix A shows that tumors in the same image are clearly visualized after injection of [^99m^Tc]Tc-DB8 with 40 and 120 μg DB8, with the upper setting of the scale adjusted to SUV 2.

**Figure 7 pharmaceutics-17-01000-f007:**
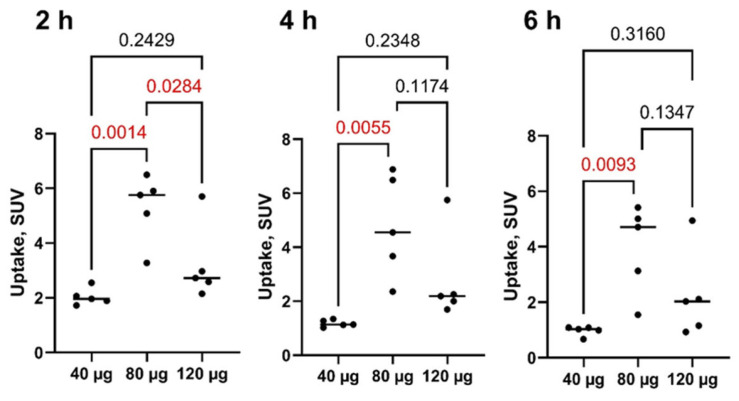
Uptake (SUV_max_) in primary ER-positive breast cancer lesions at different time points after injection of [^99m^Tc]Tc-DB8 with 40, 80, and 120 μg DB8. Statistical significance was evaluated using one-way ANOVA and is indicated in red.

**Figure 8 pharmaceutics-17-01000-f008:**
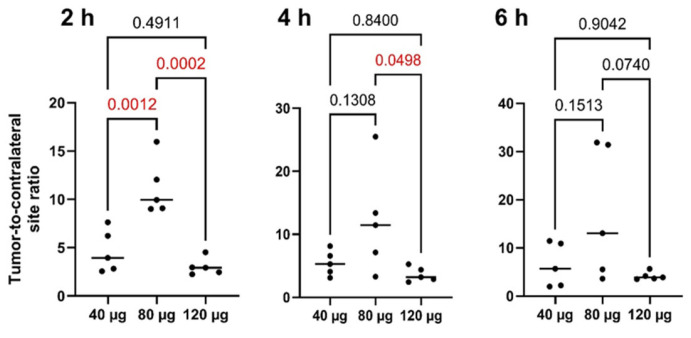
Tumor-to-contralateral breast ratios for primary ER-positive lesions at different time points after injection of [^99m^Tc]Tc-DB8 with 40, 80, and 120 μg DB8. Statistical significance was evaluated using one-way ANOVA and is indicated in red.

**Figure 9 pharmaceutics-17-01000-f009:**
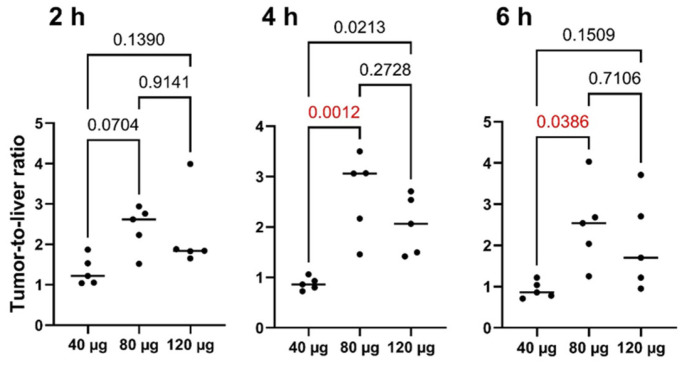
Tumor-to-liver ratios for primary ER-positive lesions at different time points after injection of [^99m^Tc]Tc-DB8 with DB8 masses of 40, 80, and 120 μg. Statistical significance was evaluated using one-way ANOVA and is indicated in red.

**Figure 10 pharmaceutics-17-01000-f010:**
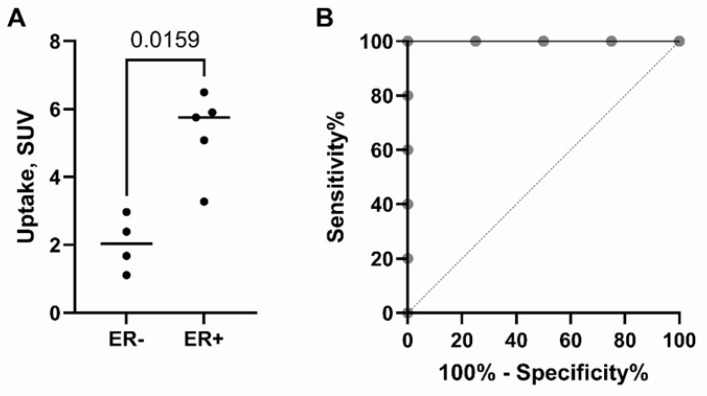
(**A**). Comparison of uptake (SUV_max_) in primary lesions of patients with ER-negative or ER-positive tumors at 2 h after injection of [^99m^Tc]Tc-DB8 with 80 μg DB8. (**B**). The receiver operating characteristic (ROC) curve analysis shows sensitivity and specificity at the SUV_max_ cut-off value 3.12 for discriminating between ER-positive and ER-negative lesions.

**Figure 11 pharmaceutics-17-01000-f011:**
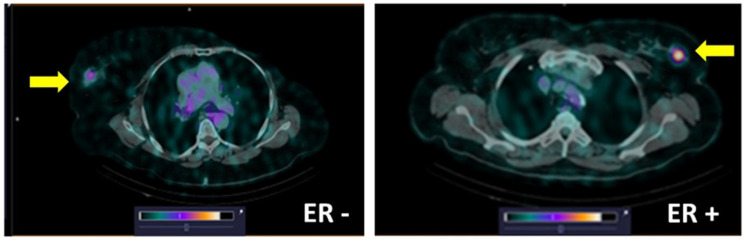
Representative SPECT/CT images of Patient 12 with a primary ER-negative tumor and Patient 8 with an ER-positive tumor 2 h after injection of [^99m^Tc]Tc-DB8 with 80 μg DB8. Arrows point at tumors. The upper setting of a linear intensity scale is adjusted to SUV 6.0 in both images.

**Figure 12 pharmaceutics-17-01000-f012:**
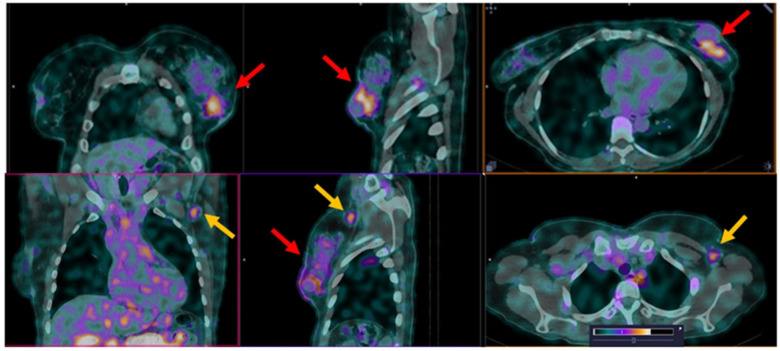
SPECT/CT images of node metastasis 2 h after injection of [^99m^Tc]Tc-DB8 (Patient 4, 120 µg DB8, 2 h after injection). Red arrows point at the primary tumor. Yellow arrows point at axillary lymph node. The upper setting of a linear intensity scale is adjusted to SUV 6.8 in all images.

**Figure 13 pharmaceutics-17-01000-f013:**
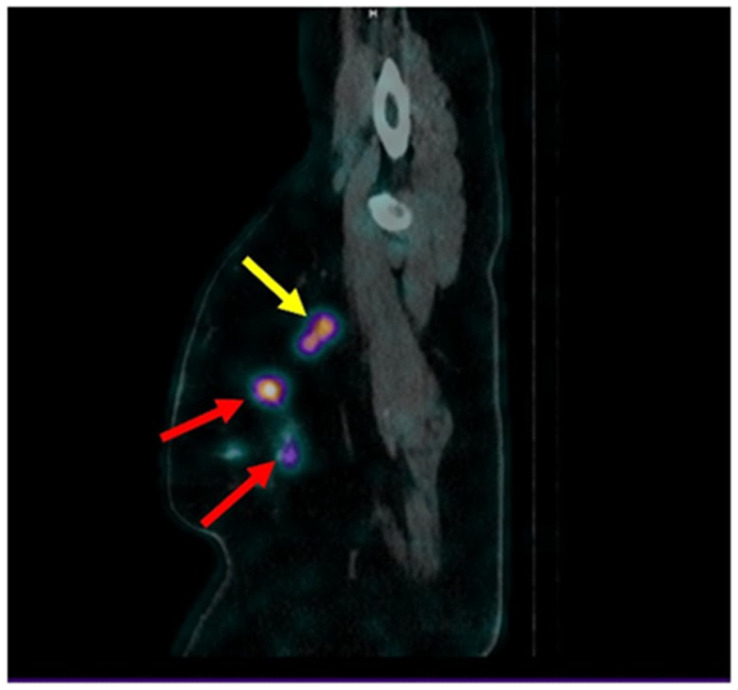
Imaging of node metastasis 2 h after injection of [^99m^Tc]Tc-DB8 (Patient 17, 120 µg DB8, 2 h after injection). Red arrows point at the multicentric primary tumor. Yellow arrow point at axillary lymph node.

**Figure 14 pharmaceutics-17-01000-f014:**
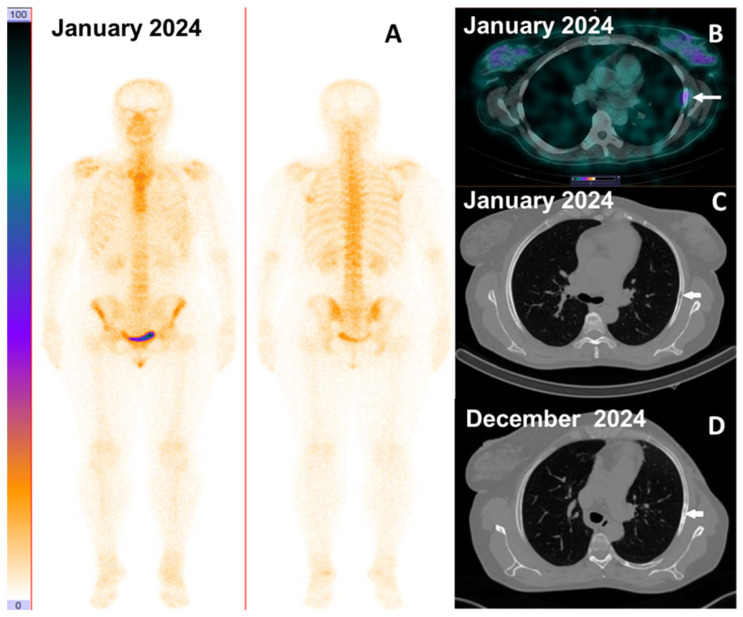
(**A**). Bone scan did not reveal bone involvement; (**B**). SPECT/CT after injection of [^99m^Tc]Tc-DB8 with 120 μg DB8 visualized abnormal accumulation in rib 5; (**C**). A site of lytic destruction (19 × 5 mm) was visualized by the chest CT; (**D**). Follow-up chest CT visualized a site of sclerotic changes (26 × 5 mm) characteristic of a response to therapy (Patient 19).

**Figure 15 pharmaceutics-17-01000-f015:**
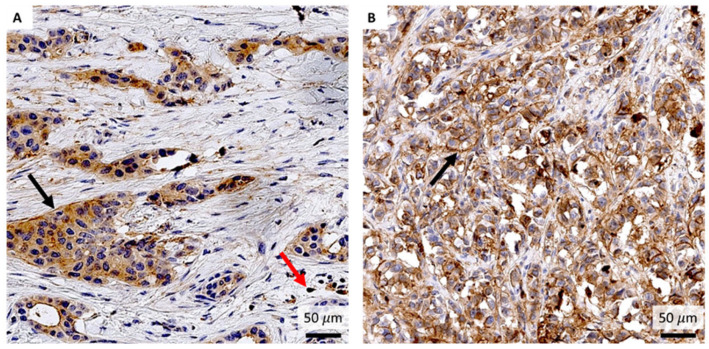
Immunohistochemical staining of ER- (**A**) and ER+ (**B**) breast cancer tissues with an anti-GRPR antibody. Black arrows indicate GRPR-positive tumor cells, while red arrows indicate GRPR-positive immune cells. Magnification: 400×.

**Figure 16 pharmaceutics-17-01000-f016:**
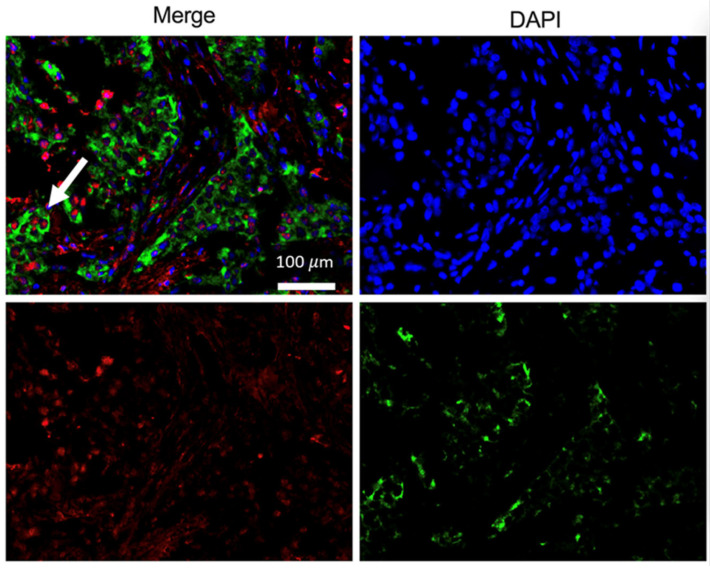
Multiplex immunofluorescent staining of ER+ breast cancer tissues with an anti-GRPR antibody (green) and anti-ER antibody (red); DAPI stains DNA to visualize nucleus (blue). White arrow indicates ER+GRPR+ tumor cells. Magnification: 400×.

**Table 1 pharmaceutics-17-01000-t001:** Patients’ characteristics before injection with [^99m^Tc]Tc-DB8.

	Age	Body Weight (kg)	Injected Mass (µg)	Clinical Stage Before Imaging	ER/PR	HER2	Ki67	Primary Tumor Size US (mm)	Injected Activity (MBq)
1	61	70	40	IIA (cT2cN0 cM0)	ER 8, PR 8	IHC 1+	8%	32 × 16 × 35	307
2	45	73.5	40	IIA (cT2cN0 cM0)	ER 8, PR 7	IHC 2+ FISH negative	45%	34 × 16 × 35	272
3	31	90.3	40	IIB (cT2cN1 cM0), multicentric growth	ER 8, PR 5	IHC 1+	25%	42 × 20 × 40	310
4	34	78.6	40	IIB (cT2cN1 cM0)	ER 8, PR 0	IHC 0	25%	28 × 20 × 33	298
5	56	64	40	IIA (cT2cN0 cM0)	ER 8, PR 7	IHC 0	15%	22 × 10 × 16	481
6	50	78.5	80	IIA (cT2cN0 cM0)	ER 8, PR 4	IHC 0	18%	24 × 17 × 25	410
7	57	74	80	IIA (cT2cN0 cM0), multicentric growth	ER 8, PR 5	IHC 1+	17%	27 × 17 × 30	560
8	71	91	80	IIA (cT2cN0 cM0)	ER 8, PR 7	IHC 1+	10%	21 × 14 × 14	465
9	54	74	80	IIA (cT2cN0 cM0)	ER 6, PR 0	IHC 1+	25–30%	32 × 16 × 21	350
10	76	67	80	IIA (cT2cN0 cM0)	ER 8, PR 6	IHC 0	3%	31 × 17 × 28	294
11	40	71	80	IIA (cT cN0 cM0)	ER 0, PR 0	IHC 0	5%	45 × 27 × 40	341
12	68	91	80	IIB (cT2cN1 cM0)	ER 0, PR 0	IHC 0	40%	29 × 17 × 25	227
13	42	63	80	IIA (cT2cN0 cM0)	ER 0, PR 0	IHC 1+	70%	23 × 15 × 30	312
14	54	77	80	IIA (cT2cN0 cM0)	ER 0, PR 0	IHC 1+	45%	25 × 16 × 23	324
15	41	44	120	IIA (cT2cN0 cM0)	ER 8, PR 8	IHC 1+	18%	16 × 23 × 25	260
16	69	54	120	IIA (cT2 N0 cM0)	ER 8, PR 7	IHC 0	7%	29 × 22 × 25	420
17	55	102	120	IIA (cT2cN0 cM0)	ER 8, PR 6	IHC 0	8%	21 × 16 × 25	355
18	49	74	120	IIA (cT2cN0 cM0)	ER 8, PR 7	IHC 1+	1%	27 × 19 × 30	367
19	50	72	120	IIIC (cT2cN3 cM0)	ER 6, PR 8	IHC 0	25%	29 × 30 × 30	448

**Table 2 pharmaceutics-17-01000-t002:** Uptake of [^99m^Tc]Tc-DB8 in organs with the highest uptake (decay-corrected). The data are presented as % IA/organ ± standard deviation.

	2 h			4 h			6 h			24 h		
	40 μg	80 μg	120 μg	40 μg	80 μg	120 μg	40 μg	80 μg	120 μg	40 μg	80 μg	120 μg
Breast	4.2 ± 1.1	6.2 ± 1.9	3.5 ± 1.1	2.4 ± 0.4	3.4 ± 1.7	2.0 ± 0.8	1.7 ± 0.3	2.7 ± 1.2	1.3 ± 0.5	0.7 ± 0.1	0.8 ± 0.4	0.6 ± 0.2
Small Intestine	4.2 ± 1.6	3.4 ± 1.1	3.3 ± 1.0	2.8 ± 1.1	2.2 ± 0.6	2.1 ± 0.7	2.1 ± 0.7	1.6 ± 0.6	1.8 ± 0.6	1.4 ± 0.3	0.7 ± 0.4	1.1 ± 0.4
Kidney	3.1 ± 1.9	5.1 ± 2.1	3.5 ± 1.6	2.1 ± 0.9	2.8 ± 0.9	2.2 ± 1.0	1.6 ± 0.5	2.2 ± 0.9	1.7 ± 0.8	0.9 ± 0.2	1.2 ± 0.8	1.1 ± 0.4
Liver	3.0 ± 1.6	3.3 ± 0.9	2.2 ± 0.9	2.1 ± 1.1	2.2 ± 0.8	1.5 ± 0.7	1.6 ± 0.8	1.4 ± 0.7	1.1 ± 0.5	0.8 ± 0.4	0.7 ± 0.3	0.5 ± 0.2
Lungs	3.8 ± 0.5	5.9 ± 1.6	3.5 ± 1.3	2.3 ± 0.3	4.1 ± 1.6	2.0 ± 1.0	1.6 ± 0.2	2.8 ± 1.2	1.6 ± 0.7	0.7 ± 0.1	0.9 ± 0.4	0.6 ± 0.2
Pancreas	1.5 ± 0.5	1.2 ± 0.3	0.8 ± 0.2 *	0.9 ± 0.4	0.8 ± 0.3	0.5 ± 0.1 *	0.7 ± 0.3	0.6 ± 0.3	0.34 ± 0.09	0.35 ± 0.09	0.2 ± 0.1	0.15 ± 0.06

* Denotes statistically significant difference (*p* < 0.05)

**Table 3 pharmaceutics-17-01000-t003:** Absorbed doses after injection of [^99m^Tc]Tc-DB8 with different DB8 masses (Data are presented as the mean mGy/MBq ± SD).

	40 µg	80 µg	120 µg
Adrenals	0.021 ± 0.003	0.025 ± 0.010	0.027 ± 0.007
Brain	0.0011 ± 0.0004	0.0012 ± 0.0004	0.0012 ± 0.0002
Breasts	0.009 ± 0.002	0.015 ± 0.009	0.007 ± 0.002
Gallbladder wall	0.007 ± 0.003	0.007 ± 0.003	0.006 ± 0.001
Lower large intestine wall	0.006 ± 0.002	0.007 ± 0.002	0.007 ± 0.002
Small intestine wall	0.007 ± 0.003	0.006 ± 0.003	0.007 ± 0.001
Stomach wall	0.006 ± 0.002	0.007 ± 0.003	0.005 ± 0.001
Upper large intestine wall	0.007 ± 0.003	0.007 ± 0.003	0.007 ± 0.001
Heart	0.004 ± 0.001	0.005 ± 0.002	0.0038 ± 0.0007
Kidneys	0.011 ± 0.005	0.014 ± 0.005	0.011 ± 0.004
Liver	0.004 ± 0.002	0.004 ± 0.001	0.004 ± 0.001
Lungs	0.005 ± 0.001	0.007 ± 0.002	0.005 ± 0.001
Muscle	0.0020 ± 0.0008	0.0022 ± 0.0007	0.0022 ± 0.0005
Ovaries	0.019 ± 0.005	0.022 ± 0.007	0.028 ± 0.009
Pancreas	0.016 ± 0.005	0.012 ± 0.004	0.009 ± 0.002 *
Red marrow	0.003 ± 0.001	0.003 ± 0.001	0.0029 ± 0.0005
Osteogenic cells	0.006 ± 0.003	0.007 ± 0.003	0.006 ± 0.001
Skin	0.0014 ± 0.0007	0.0016 ± 0.0006	0.0017 ± 0.0004
Spleen	0.004 ± 0.001	0.005 ± 0.001	0.005 ± 0.001
Thymus	0.008 ± 0.001	0.010 ± 0.004	0.009 ± 0.002
Thyroid	0.017 ± 0.003	0.023 ± 0.007	0.018 ± 0.007
Urinary bladder wall	0.02 ± 0.01	0.02 ± 0.01	0.02 ± 0.01
Uterus	0.010 ± 0.002	0.009 ± 0.001	0.010 ± 0.003
Effective Dose Equivalent (mSv/MBq)	0.012 ± 0.001	0.015 ± 0.007	0.019 ± 0.006
Effective Dose (mSv/MBq)	0.009 ± 0.001	0.011 ± 0.004	0.014 ± 0.003 *

* Significant difference after injection of 40 and 120 μg DB8.

## Data Availability

Data are contained within the article and in the Appendix A.

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
