# Peer review of "The Impact of the Injected Mass of the Gastrin-Releasing Peptide Receptor Antagonist on Uptake in Breast Cancer: Lessons from a Phase I Trial of [99mTc]Tc-DB8"

_pharmaceutics, 2025, doi:10.3390/pharmaceutics17081000_

Round 1
Reviewer 1 Report
Comments and Suggestions for Authors
This manuscript entitled “The Impact of the Injected Mass of the Gastrin-Releasing Peptide Receptor Antagonist on Uptake in Breast Cancer: Lessons from a Phase I trial of [99mTc]Tc-DB8” described the investigation of the [99mTc]Tc-DB8 injection mass on ER-positive breast cancer patients. The authors found that [99mTc]Tc-DB8 (with up to 120 μg DB8) was well tolerated, and the optimal injection mass of DB8 was ~80 μg for clinical [99mTc]Tc-DB8 scan. The manuscript was well organized and presented, and the significance of the proposed study is high. Below are some specific comments:
1) It is well known that the mass of injected radiotracer could affect its clinical performance (especially when the total amount of targeted receptor in body is not very low), and thus the “Novelty” of this study is "Average".
2) The reference “28” (Line 132) concluded that [99mTc]DB15 (with Sar11) showed attractive pharmacokinetic profile over the one with Gly11, and its further clinical evaluation seemed to be warranted. What are the advantage(s) of using [99mTc]DB8 with Gly11 (not [99mTc]DB15 with Sar11) for this clinical trial?
3) The body weight is another important factor to determine the injection mass, and thus please provide the weight of each patient; “μg DB8/kg” may be more appropriate to be used to study the injection mass.
4) Would suggest the authors to study the correlations between tumor SUVs obtained from SPECT scan and GRPR expression obtained from ex vivo test. As the observed higher tumor SUVs might be thought to be result of higher GRPR expression, instead of injection mass.
Author Response
Reviewer 1#
Open Review
(x) I would not like to sign my review report
( ) I would like to sign my review report
Quality of English Language
( ) The English could be improved to more clearly express the research.
(x) The English is fine and does not require any improvement.
Yes Can be improved Must be improved Not applicable
Does the introduction provide sufficient background and include all relevant references?
(x) ( ) ( ) ( )
Is the research design appropriate?
( ) (x) ( ) ( )
Are the methods adequately described?
( ) (x) ( ) ( )
Are the results clearly presented?
(x) ( ) ( ) ( )
Are the conclusions supported by the results?
(x) ( ) ( ) ( )
Are all figures and tables clear and well-presented?
(x) ( ) ( ) ( )
Comments and Suggestions for Authors
This manuscript entitled “The Impact of the Injected Mass of the Gastrin-Releasing Peptide Receptor Antagonist on Uptake in Breast Cancer: Lessons from a Phase I trial of [99mTc]Tc-DB8” described the investigation of the [99mTc]Tc-DB8 injection mass on ER-positive breast cancer patients. The authors found that [99mTc]Tc-DB8 (with up to 120 μg DB8) was well tolerated, and the optimal injection mass of DB8 was ~80 μg for clinical [99mTc]Tc-DB8 scan. The manuscript was well organized and presented, and the significance of the proposed study is high. Below are some specific comments:
Answer: Thank you very much for the high appreciation of our work.
- It is well known that the mass of injected radiotracer could affect its clinical performance (especially when the total amount of targeted receptor in body is not very low), and thus the “Novelty” of this study is "Average".
Answer: We agree with the reviewer. We pointed out in the Introduction that previous studies demonstrated the importance of the selection of the optimal injected mass of the radiotracer. Furthermore, we pointed out in the Discussion “In principle, this agreed with the data obtained for different classes of peptide-based radionuclide imaging probes. Kooij and co-authors [49] published in 1994 an abstract reporting that injection of 1 µg of [111In]In-Octreoscan (somatostatin receptor agonist) resulted in poor detection of neuroendocrine tumors and injection of [111In]In-Octreoscan containing 5-200 µg unlabeled DTPA-octreotide provided better visualization. The full paper was not published, but this information was repeated in papers reporting preclinical investigations of this issue [50, 51]. The data from rat models confirmed that both tumor uptake and tumor-to-organ ratios were the highest not in the case of the lowest injected mass. A reduction of tracer uptake in normal somatostatin receptor-expressing organs leading to a decrease of background and making it more available for tumors was proposed as an explanation for this phenomenon. Similar phenomena were observed in clinics for the 68Ga-labeled somatostatin receptor agonist DOTATOC [52] but not for the antagonist OPS202 [53]. It should be however emphasized that all this data has been acquired from clinical (and preclinical) studies with radiolabeled SST2R-agonists, while our study has been exclusively focused on a [99mTc]Tc-labeled GRPR-antagonist. Such study in human is performed for the first time and was possible and safe to conduct due to the lack of adverse effects triggered by binding of the antagonist to the GRPR. It thereby represents the first paradigm for further studies and practices with this class of compounds, potentially significant for enhancing their theranostic value.
- The reference “28” (Line 132) concluded that [99mTc]DB15 (with Sar11) showed attractive pharmacokinetic profile over the one with Gly11, and its further clinical evaluation seemed to be warranted. What are the advantage(s) of using [99mTc]DB8 with Gly11 (not [99mTc]DB15 with Sar11) for this clinical trial?
Answer: The [99mTc]Tc-DB8 tracer was selected based on the results of a comparative preclinical evaluation study, for showing highest uptake and retention in a human PC-3 prostate cancer mice model (ref. 27 JMEDCHEM). The decision was subsequently made to see how this tracer performs in human in BCa patients for the reasons outlined in the Introduction.
Concurrently, the [99mTc]Tc-DB15 tracer developed later, showed higher in vivo stability in mice by virtue of the Gly11 to Sar11 substitution. [99mTc]Tc-DB15 showed tumor uptake and retention in mice with both PC-3 and breast cancer xenografts in mice and was later studied in a very limited number of very advanced BCa patients in another clinical center.
The overall organization of the present study has been much different, with a different clinical center involved, selection of patients being much stricter and more systematic, including the consideration of important biochemical traits of the disease. Organization means, approval of protocols, ordering and delivery of DB8 and patient recruitment, all of which take considerable time to be completed.
So, the first clinical evaluation of these two tracers has been developing in parallel. Moreover, one cannot be sure about the superiority” of [99mTc]Tc-DB15 over [99mTc]Tc-DB8 in a clinical setting, given that too many parameters may affect outcomes. In the present study, the use of different peptide mass alone was shown to already have an effect. On the other hand, translation of preclinical results in a 25 g mouse to an average 60 kg body female patient may not be straightforward either. There are differences in the pattern of GRPR expression in mice and humans, pharmacokinetic factors, interspecies GRPR, enzymatic activity rates, cardiac rhythm, and other differences.
We kindly invite the Reviewer to: De Jong M, Maina T. Of mice and humans: Are they the same? – Implications in cancer translational research. J. Nucl. Med. 51(4): 501-504; 2010; DOI https://doi.org/10.2967/jnumed.109.065706 .
A direct comparison of the two tracers has not been conducted even at the preclinical level. However, by comparing the retention in the PC-3 tumor mice model, we cannot see a clear different impact of the 15% stability improvement of [99mTc]Tc-DB15 (75% stable at 5 min pi) on tumor retention compared with [99mTc]Tc-DB8 (60% stable at 5 min pi). E.g. [99mTc]Tc-DB15 showed 17.79 ± 1.58 %IA/g, while [99mTc]Tc-DB8 16.32 ± 1.82 %IA/g tumor uptake at 24 h pi.
- The body weight is another important factor to determine the injection mass, and thus please provide the weight of each patient; “μg DB8/kg” may be more appropriate to be used to study the injection mass.
Answer: Thank you for pointing out this important aspect. To address this comment, we provided body weight for each patient in Table 1. We have also provided Supplementary Figure 1, showing that there was no significant difference (p >0.05, one-way ANOVA test) between groups, and added the following sentence to the Subsection 2.1. Patients “There was no significant difference between body weight of patient’s cohorts (Supplementary Figure 1)”.
- Would suggest the authors to study the correlations between tumor SUVs obtained from SPECT scan and GRPR expression obtained from ex vivo test. As the observed higher tumor SUVs might be thought to be result of higher GRPR expression, instead of injection mass.
Answer: We plan to study such correlation in a planned clinical study on extended cohorts of patients. One of the issues is that there is no internationally recognized scale for evaluation of GRPR expression. For the moment, we validate available antibodies to GRPR and reproducibility of staining. We intend also to apply for an ethical permit to perform imaging in the same patients twice, with higher and lower injected mass.
Reviewer 2 Report
Comments and Suggestions for Authors
The manuscript by Bragina et al is interesting and written well. The English is clear and understandable.
The radiochemistry is carried out clearly using standard TcO4- conditions, with good RCY. The addition of the cold peptide is not fully clear, as it's not clear why one would vary the method if there is a 'best' method, or indeed, a lowest dose method in terms of the patient and any toxicity.
With respect to the clinical imaging itself, the biggest issue is the number of patients, 5 for 40 and 120 ug, and then 10 for 80 ug, is low, to the point that statistics and clear conclusions are difficult. As a result, I would argue the discussion and conclusions should be limited at best. Furthermore, once sex as a variable is added in, along with age of the patient, it makes the study close to not relevant. A minor note - the sex of the patients isn't clear in the text, to make it clear how many are in each group. Linking back to the radiochemistry, one wonders if some of the discoveries linked between cold compound and the tumor uptake could have been worked through in a preclinical setting, rather than a clinical experiment. One last note on the dosimetry, if one has this right, 120 ug is definitely worse (although not dangerous) in terms of effective dose to a patient, though the authors do not really discuss this or talk about it.
Another minor note, a lot of the images and the peptide in Figure 1 or Figure 8 look very fuzzy and need to be fixed before.
I think that the discussion and conclusion need reworking before the paper is accepted.
Author Response
Reviewer 2
Open Review
(x) I would not like to sign my review report
( ) I would like to sign my review report
Quality of English Language
( ) The English could be improved to more clearly express the research.
(x) The English is fine and does not require any improvement.
Yes Can be improved Must be improved Not applicable
Does the introduction provide sufficient background and include all relevant references?
(x) ( ) ( ) ( )
Is the research design appropriate?
(x) ( ) ( ) ( )
Are the methods adequately described?
( ) (x) ( ) ( )
Are the results clearly presented?
(x) ( ) ( ) ( )
Are the conclusions supported by the results?
( ) ( ) (x) ( )
Are all figures and tables clear and well-presented?
( ) (x) ( ) ( )
Comments and Suggestions for Authors
The manuscript by Bragina et al is interesting and written well. The English is clear and understandable.
We thank the Reviewer for these positive comments.
The radiochemistry is carried out clearly using standard TcO4- conditions, with good RCY. The addition of the cold peptide is not fully clear, as it's not clear why one would vary the method if there is a 'best' method, or indeed, a lowest dose method in terms of the patient and any toxicity.
Answer: We would like to draw the Reviewer’s attention to the fact that the addition of “cold” DB8 was intended given that a major objective of this study was to evaluate the effect of peptide mass on imaging quality with SPECT/CT. It should be emphasized that DB8 is a GRPR-antagonist, not activating the GRPR upon binding and therefore, safe for injection in higher mass to patients.
With respect to the clinical imaging itself, the biggest issue is the number of patients, 5 for 40 and 120 ug, and then 10 for 80 ug, is low, to the point that statistics and clear conclusions are difficult. As a result, I would argue the discussion and conclusions should be limited at best.
Answer: We understand that the sample size is limited. As we wrote in the Discussion “An obvious limitation of this study is the small patient population typical for phase I trials. Thus, the statistical power is insufficient for strong statements and predictions. “. However, the statistical treatment results suggest that the difference is pronounced enough to find a significant difference.
It should be noted that early clinical studies are typically performed in limited cohorts of patients, but might provide important and quite strong conclusions. Below we list just a few examples for papers published in the Journal of Nuclear Medicine. The Journal is a leading publishing platform in the field of nuclear medicine and known by meticulous and stringent peer-reviewing. We do not feel that our conclusions are too strong compared with these studies.
- Ichijo S, Arisawa T, Hatano M, Nakajima W, Miyazaki T, Eiro T, Takada Y, Iai R, Sano A, Sonoda M, Takayama Y, Kimura Y, Takahashi T. First-in-Human Study of 18F-Labeled PET Tracer for Glutamate AMPA Receptor [18F]K-40: A Derivative of [11C]K-2. J Nucl Med. 2025 Jun 2;66(6):932-939.
Five healthy volunteers were enrolled in this study. The conclusion was “ Based on the evidence described in this study, [18F]K-40 is an excellent AMPAR PET drug for AMPAR quantification without blood sampling”
- Gondry O, Xavier C, Raes L, Heemskerk J, Devoogdt N, Everaert H, Breckpot K, Lecocq Q, Decoster L, Fontaine C, Schallier D, Aspeslagh S, Vaneycken I, Raes G, Van Ginderachter JA, Lahoutte T, Caveliers V, Keyaerts M. Phase I Study of [68Ga]Ga-Anti-CD206-sdAb for PET/CT Assessment of Protumorigenic Macrophage Presence in Solid Tumors (MMR Phase I). J Nucl Med. 2023 Sep;64(9):1378-1384.
Seven patients were enrolled in this study. Conclusion was “[68Ga]Ga-NOTA-anti-CD206-sdAb is safe and well tolerated. It shows rapid blood clearance and renal excretion, enabling high contrast-to-noise imaging at 90 min after injection. The radiation dose is comparable to that of routinely used PET tracers. “
- Cheng K, Ge L, Song M, Li W, Zheng J, Liu J, Luo Y, Sun P, Xu S, Cheng Z, Yu J, Liu J. Preclinical Evaluation and Pilot Clinical Study of CD137 PET Radiotracer for Noninvasive Monitoring Early Responses of Immunotherapy. J Nucl Med. 2025 Jan 3;66(1):40-46.
Five patients diagnosed with hepatocellular carcinoma were enrolled in this study. The conclusion was “We demonstrated the utility of [18F]AlF-NOTA-BCP137 PET imaging in the assessment of CD137 expression, and our findings revealed the potential of this imaging method for the early noninvasive evaluation of activated T cells and tumor responses to immunotherapy”
- von Guggenberg E, di Santo G, Uprimny C, Bayerschmidt S, Warwitz B, Hörmann AA, Zavvar TS, Rangger C, Decristoforo C, Sviridenko A, Nilica B, Santo G, Virgolini IJ. Safety, Biodistribution, and Radiation Dosimetry of the 68Ga-Labeled Minigastrin Analog DOTA-MGS5 in Patients with Advanced Medullary Thyroid Cancer and Other Neuroendocrine Tumors. J Nucl Med. 2025 Feb 3;66(2):257-263.
Six patients with advanced MTC and 6 patients with gastroenteropancreatic and bronchopulmonary NETs were enrolled. The conclusion was “Besides confirming the safety of administration, within the phase I part of the prospective clinical trial, an acceptable effective whole-body dose, an overall favorable biodistribution, and the feasibility of cholecystokinin-2 receptor imaging could be shown for 68Ga-DOTA-MGS5”
- Li L, Lin X, Wang L, Ma X, Zeng Z, Liu F, Jia B, Zhu H, Wu A, Yang Z. Immuno-PET of colorectal cancer with a CEA-targeted [68Ga]Ga-nanobody: from bench to bedside. Eur J Nucl Med Mol Imaging. 2023 Oct;50(12):3735-3749.
Phase I study was conducted on 9 patients with primary and metastatic CRC. The conclusion was “[68Ga]Ga-HNI01 is a novel CEA-targeted PET imaging radiotracer with excellent pharmacokinetics and favorable dosimetry profiles.”
- Gillett D, Senanayake R, MacFarlane J, Bashari W, Palma A, Hu L, Harper I, Mendichovszky IA, Antoni G, Hellman P, Sundin A, Hird M, Boros I, Brown MJ, Cheow H, Aloj L, Aigbirhio F, Gurnell M. A Phase I/IIa Clinical Trial to Evaluate Safety and Adrenal Uptake of Para-Chloro-2-[18F]Fluoroethyletomidate in Healthy Volunteers and Patients with Primary Aldosteronism. J Nucl Med. 2025 Mar 3;66(3):434-440.
The phase I was performed on 6 patients with PA (3 unilateral disease, 3 bilateral disease) and 5 healthy volunteers. The conclusion was “Distinction between APAs and normal adrenal tissue is enhanced by dexamethasone pretreatment to suppress [18F]CETO uptake by normal glands. This positions [18F]CETO as a promising imaging tool for evaluation in the context of PA”
Furthermore, once sex as a variable is added in, along with age of the patient, it makes the study close to not relevant. A minor note - the sex of the patients isn't clear in the text, to make it clear how many are in each group.
Answer: Thank you for pointing this out. We considered that by default, an imaging of breast cancer would indicate female patients. Furthermore, we wrote in the Discussion part ”The aim was to assess the performance in both breast and prostate cancers. Taking into account that the previous study showed a noticeable difference in the biodistribution and dosimetry of [99mTc]Tc-maSSS-RM26 in male and female patients [32], most likely due to gender-related differences in body size and physiology, the data are presented separately. In the case of breast cancer, this Phase I study was performed in patients with primary tumors, with potential lymph node involvement. “However, there is some incidence of breast cancer in males. To eliminate any uncertainty, we have modified the text in the Abstract as: Fifteen female patients with ER-positive primary breast cancer were enrolled and divided into three cohorts… “ and in the Materials and Methods section, Patients subsection “Nineteen female patients were enrolled (Table 1; Figure 2)” and stressed additionally in the Discussion section “In the case of breast cancer, this Phase I study was performed in female patients with primary tumors, with potential lymph node involvement“.
Linking back to the radiochemistry, one wonders if some of the discoveries linked between cold compound and the tumor uptake could have been worked through in a preclinical setting, rather than a clinical experiment.
Answer: Thank you for pointing out this important question. We have missed references to preclinical studies. To address the comment, we have modified the text in the Discussion section in the following way: “The tumor uptake (2 h post-injection) was significantly higher after injection of 80 µg DB8. The increase of SUVmax was more than twofold compared to the value after injection of 40 µg. Two-fold was also The twofold difference in tumor-to-contralateral breast ratio was also observed in the clinical study. The further increase of the injection mass to 120 µg did not improve the tumor uptake or tumor-to-organ ratios, presumably due to the partial saturation of binding sites in tumors by the cold DB8. This finding is consistent with preclinical data obtained in mice for the agonistic bombesin analogue [67Ga]Ga-DOTA-BZH3 [49] and antagonistic [111In]In-NOTA-P2-RM26 [50]. In both cases, injection of 15 pmol per mouse resulted in significantly higher tumour uptake compared with injections of smaller or larger masses. However, the effect was not as strong as in this clinical study; the increase was between 20 and 56% for the antagonist and about 40% for the agonist. “
- Schuhmacher, J.; Zhang, H.; Doll, J.; Mäcke, H. R.; Matys, R.; Hauser, H.; Henze, M.; Haberkorn, U.; Eisenhut, M. GRP receptor-targeted PET of a rat pancreas carcinoma xenograft in nude mice with a 68Ga-labeled bombesin(6-14) analog. J. Nuc.l Med. 2005, 46, 691-699.
- Varasteh, Z.; Velikyan, I.; Lindeberg, G.; Sörensen, J.; Larhed, M.; Sandström, M.; Selvaraju, R. K.; Malmberg, J.; Tolmachev, V.; Orlova, A. Synthesis and characterization of a high-affinity NOTA-conjugated bombesin antagonist for GRPR-targeted tumor imaging. Bioconjg. Chem., 203, 24, 1144–1153. https://doi.org/10.1021/bc300659k
One should consider that women patients (Homo Sapiens) of average 60-kg body weight may display significantly divergent behaviour compared with mice (Mus Musculus) of average 25 g body weight. Therefore, conclusions drawn from results of a preclinical mass escalation study in mice cannot be safely extrapolated to the human situation. First of all, there is obvious differences in the pattern of GRPR expression between mice and humans, but also in the affinity of the tracer between species. Further discrepancies may arise from different pharmacokinetic factors, differences in enzymatic activity (NEP local concentrations and types), cardiac rhythm etc, etc, etc.
We invite the Reviewer to consider a JNM paper commenting on these issues: De Jong M, Maina T. Of mice and humans: Are they the same? – Implications in cancer translational research. J. Nucl. Med. 51(4): 501-504; 2010; DOI https://doi.org/10.2967/jnumed.109.065706 .
Therefore, only a clinical study will reliably and unequivocally address a clinical question, although important information from animal studies can be helpful.
One last note on the dosimetry, if one has this right, 120 ug is definitely worse (although not dangerous) in terms of effective dose to a patient, though the authors do not really discuss this or talk about it.
Answer: Thank you for pointing this out. To address this comment, we have added the following text to the conclusion paragraph of the Discussion section “The dosimetry after injection of 120 µg DB8 was definitely worse in terms of effective dose to a patient, although still safe. “.
Another minor note, a lot of the images and the peptide in Figure 1 or Figure 8 look very fuzzy and need to be fixed before.
Answer: Thank you for pointing this out. The Figures look really blurry in the PDF file for peer-reviewing. I can assure that the images are much clearer in the Word.docxs version, which we received for the revision. In addition, we have submitted a Zip-File with the high-resolution images. I hope that these files will be used for the final preparation of manuscript by the Editorial Office,
I think that the discussion and conclusion need reworking before the paper is accepted.
Answer: A few critical comments addressing these concerns have been now included in the revised version of the main article and Supplementary Files.
Reviewer 3 Report
Comments and Suggestions for Authors
This manuscript “The Impact of the Injected Mass of the Gastrin-Releasing Peptide Receptor Antagonist on Uptake in Breast Cancer: Lessons from a Phase I trial of [99mTc]Tc-DB8” presents an interesting study on human trial of [99mTc]Tc-DB8 GRPR-antagonistic peptide on safety, biodistribution, and dosimetry. The study reports interesting results that there were no adverse side effects associated with [99mTc]Tc-DB8 injections. The effective dose of [99mTc]Tc-DB8 was 0.009-0.014 mSv/MBq. The injection of an optimal mass (80 μg) provides the highest uptake in ER-positive tumors. Primary tumors and all known lymph node metastases were visualized irrespective of injected peptide mass.
- The study ultimately involved 19 participants, a sample size that is limited for reaching definitive conclusions. Authors are required to justify this.
- Table 3; The authors should present this data as a percentage of the administered dose, as this format provides better clarity and facilitates more meaningful correlations with dose than expressing it in mean mGy/MBq ± SD.
- Fig 6; The images lack clarity, and the tumor along with the reference phantoms are not clearly distinguishable. Additionally, each image should be labeled with the corresponding patient ID for better identification and reference.
- What could be the reason for the significantly lower uptake at the 120 µg dose compared to 40 µg? Do the authors propose that further dose optimization in the range of 80 µg to 120 µg might be beneficial?"
- Line 375-378; “There was no significant difference (p = 0.3, paired t-test) between tumor uptake at 2 h (SUVmax = 5.3 ± 1.4) and 4 h (SUVmax = 4.8 ± 1.9) after injection of [99mTc]Tc-DB8 with 80 μg DB8, but the uptake at 6 h after injection (SUV max = 4.0 ± 1.6) was significantly lower.” This interpretation does not corroborate with Fig 7.
Author Response
Reviewer 3
Open Review
(x) I would not like to sign my review report
( ) I would like to sign my review report
Quality of English Language
( ) The English could be improved to more clearly express the research.
(x) The English is fine and does not require any improvement.
Yes Can be improved Must be improved Not applicable
Does the introduction provide sufficient background and include all relevant references?
(x) ( ) ( ) ( )
Is the research design appropriate?
(x) ( ) ( ) ( )
Are the methods adequately described?
(x) ( ) ( ) ( )
Are the results clearly presented?
(x) ( ) ( ) ( )
Are the conclusions supported by the results?
(x) ( ) ( ) ( )
Are all figures and tables clear and well-presented?
( ) ( ) (x) ( )
Comments and Suggestions for Authors
This manuscript “The Impact of the Injected Mass of the Gastrin-Releasing Peptide Receptor Antagonist on Uptake in Breast Cancer: Lessons from a Phase I trial of [99mTc]Tc-DB8” presents an interesting study on human trial of [99mTc]Tc-DB8 GRPR-antagonistic peptide on safety, biodistribution, and dosimetry. The study reports interesting results that there were no adverse side effects associated with [99mTc]Tc-DB8 injections. The effective dose of [99mTc]Tc-DB8 was 0.009-0.014 mSv/MBq. The injection of an optimal mass (80 μg) provides the highest uptake in ER-positive tumors. Primary tumors and all known lymph node metastases were visualized irrespective of injected peptide mass.
- The study ultimately involved 19 participants, a sample size that is limited for reaching definitive conclusions. Authors are required to justify this.
Answer: We understand that the sample size is limited. As we wrote in the Discussion “An obvious limitation of this study is the small patient population typical for phase I trials. Thus, the statistical power is insufficient for strong statements and predictions. “. However, the statistical treatment results suggest that the difference is pronounced enough to find a significant difference.
It has to be noted that early clinical studies are typically performed in limited cohorts of patients, but might provide strong and important and quite strong conclusions. Below we list just a few examples for papers published in the Journal of Nuclear Medicine. The Journal is a leading publishing platform in the field of nuclear medicine and known by meticulous and stringent peer-reviewing. We do not feel that our conclusions are too strong compared with these studies.
- Ichijo S, Arisawa T, Hatano M, Nakajima W, Miyazaki T, Eiro T, Takada Y, Iai R, Sano A, Sonoda M, Takayama Y, Kimura Y, Takahashi T. First-in-Human Study of 18F-Labeled PET Tracer for Glutamate AMPA Receptor [18F]K-40: A Derivative of [11C]K-2. J Nucl Med. 2025 Jun 2;66(6):932-939.
Five healthy volunteers were enrolled in this study. The conclusion was “ Based on the evidence described in this study, [18F]K-40 is an excellent AMPAR PET drug for AMPAR quantification without blood sampling”
- Gondry O, Xavier C, Raes L, Heemskerk J, Devoogdt N, Everaert H, Breckpot K, Lecocq Q, Decoster L, Fontaine C, Schallier D, Aspeslagh S, Vaneycken I, Raes G, Van Ginderachter JA, Lahoutte T, Caveliers V, Keyaerts M. Phase I Study of [68Ga]Ga-Anti-CD206-sdAb for PET/CT Assessment of Protumorigenic Macrophage Presence in Solid Tumors (MMR Phase I). J Nucl Med. 2023 Sep;64(9):1378-1384.
Seven patients were enrolled in this study. Conclusion was “[68Ga]Ga-NOTA-anti-CD206-sdAb is safe and well tolerated. It shows rapid blood clearance and renal excretion, enabling high contrast-to-noise imaging at 90 min after injection. The radiation dose is comparable to that of routinely used PET tracers. “
- Cheng K, Ge L, Song M, Li W, Zheng J, Liu J, Luo Y, Sun P, Xu S, Cheng Z, Yu J, Liu J. Preclinical Evaluation and Pilot Clinical Study of CD137 PET Radiotracer for Noninvasive Monitoring Early Responses of Immunotherapy. J Nucl Med. 2025 Jan 3;66(1):40-46.
Five patients diagnosed with hepatocellular carcinoma were enrolled in this study. The conclusion was “We demonstrated the utility of [18F]AlF-NOTA-BCP137 PET imaging in the assessment of CD137 expression, and our findings revealed the potential of this imaging method for the early noninvasive evaluation of activated T cells and tumor responses to immunotherapy”
- von Guggenberg E, di Santo G, Uprimny C, Bayerschmidt S, Warwitz B, Hörmann AA, Zavvar TS, Rangger C, Decristoforo C, Sviridenko A, Nilica B, Santo G, Virgolini IJ. Safety, Biodistribution, and Radiation Dosimetry of the 68Ga-Labeled Minigastrin Analog DOTA-MGS5 in Patients with Advanced Medullary Thyroid Cancer and Other Neuroendocrine Tumors. J Nucl Med. 2025 Feb 3;66(2):257-263.
Six patients with advanced MTC and 6 patients with gastroenteropancreatic and bronchopulmonary NETs were enrolled. The conclusion was “Besides confirming the safety of administration, within the phase I part of the prospective clinical trial, an acceptable effective whole-body dose, an overall favorable biodistribution, and the feasibility of cholecystokinin-2 receptor imaging could be shown for 68Ga-DOTA-MGS5”
- Li L, Lin X, Wang L, Ma X, Zeng Z, Liu F, Jia B, Zhu H, Wu A, Yang Z. Immuno-PET of colorectal cancer with a CEA-targeted [68Ga]Ga-nanobody: from bench to bedside. Eur J Nucl Med Mol Imaging. 2023 Oct;50(12):3735-3749.
Phase I study was conducted on 9 patients with primary and metastatic CRC. The conclusion was “[68Ga]Ga-HNI01 is a novel CEA-targeted PET imaging radiotracer with excellent pharmacokinetics and favorable dosimetry profiles.”
- Gillett D, Senanayake R, MacFarlane J, Bashari W, Palma A, Hu L, Harper I, Mendichovszky IA, Antoni G, Hellman P, Sundin A, Hird M, Boros I, Brown MJ, Cheow H, Aloj L, Aigbirhio F, Gurnell M. A Phase I/IIa Clinical Trial to Evaluate Safety and Adrenal Uptake of Para-Chloro-2-[18F]Fluoroethyletomidate in Healthy Volunteers and Patients with Primary Aldosteronism. J Nucl Med. 2025 Mar 3;66(3):434-440.
The phase I was performed on 6 patients with PA (3 unilateral disease, 3 bilateral disease) and 5 healthy volunteers. The conclusion was “Distinction between APAs and normal adrenal tissue is enhanced by dexamethasone pretreatment to suppress [18F]CETO uptake by normal glands. This positions [18F]CETO as a promising imaging tool for evaluation in the context of PA”
- Table 3; The authors should present this data as a percentage of the administered dose, as this format provides better clarity and facilitates more meaningful correlations with dose than expressing it in mean mGy/MBq ± SD.
Answer: Table 3 presents the dosimetry calculation results. For currently applied regulation see e.g. Stokke C, Gnesin S, Tran-Gia J, Cicone F, Holm S, Cremonesi M, Blakkisrud J, Wendler T, Gillings N, Herrmann K, Mottaghy FM, Gear J. EANM guidance document: dosimetry for first-in-human studies and early phase clinical trials. Eur J Nucl Med Mol Imaging. 2024 “Apr;51(5):1268-1286: “Absorbed radiation dose estimates shall be calculated according to a specified, internationally recognized system by a particular route of administration “and “The absorbed dose is the energy absorbed per unit mass, and its unit is joule per kilogram (J/kg), which is given the name gray (Gy)….The unit for the effective dose is the same as for absorbed and equivalent dose, J/kg, and is named sievert (Sv)”. Thus, our data are presented in accordance with the currently valid guidelines.
In the recent papers in the Journal of Nuclear Medicine or European Journal of Nuclear Medicine and Molecular Imaging, the dosimetry is reported as mGy/MBq for absorbed doses and as mSv/MBq for effective dose in the following examples:
- Gondry O, Xavier C, Raes L, Heemskerk J, Devoogdt N, Everaert H, Breckpot K, Lecocq Q, Decoster L, Fontaine C, Schallier D, Aspeslagh S, Vaneycken I, Raes G, Van Ginderachter JA, Lahoutte T, Caveliers V, Keyaerts M. Phase I Study of [68Ga]Ga-Anti-CD206-sdAb for PET/CT Assessment of Protumorigenic Macrophage Presence in Solid Tumors (MMR Phase I). J Nucl Med. 2023 Sep;64(9):1378-1384.
- von Guggenberg E, di Santo G, Uprimny C, Bayerschmidt S, Warwitz B, Hörmann AA, Zavvar TS, Rangger C, Decristoforo C, Sviridenko A, Nilica B, Santo G, Virgolini IJ. Safety, Biodistribution, and Radiation Dosimetry of the 68Ga-Labeled Minigastrin Analog DOTA-MGS5 in Patients with Advanced Medullary Thyroid Cancer and Other Neuroendocrine Tumors. J Nucl Med. 2025 Feb 3;66(2):257-263.
- Li L, Lin X, Wang L, Ma X, Zeng Z, Liu F, Jia B, Zhu H, Wu A, Yang Z. Immuno-PET of colorectal cancer with a CEA-targeted [68Ga]Ga-nanobody: from bench to bedside. Eur J Nucl Med Mol Imaging. 2023 Oct;50(12):3735-3749.
- Lindenberg L, Hope TA, Lin FI, Rowe SP, Pucar D, Gilbert N, Chicco D, He B, Feuerecker B, Castaldi E, Solnes LB. Safety, Dosimetry, and Feasibility of [68Ga]Ga-PSMA-R2 as an Imaging Agent in Patients with Biochemical Recurrence or Metastatic Prostate Cancer. J Nucl Med. 2025 Mar 3;66(3):359-365.
Thus, our format of reporting dosimetry does not deviate from high international standards.
Furthermore, the information on the uptake as percentage of injected activity is provided in the Table 2:
“Table 2. Uptake of [99mTc]Tc-DB8 in organs with the highest uptake (decay-corrected). The data are presented as % IA/organ ± standard deviation.” Since we measure activity in the clinical study, we use the term “% injected activity (IA)/organ and ”not “ injected dose/ Organ” to avoid confusion with absorbed doses.
- Fig 6; The images lack clarity, and the tumor along with the reference phantoms are not clearly distinguishable. Additionally, each image should be labeled with the corresponding patient ID for better identification and reference.
Answer: Thank you for pointing out a number of issues.
The patients ID issue. Patients ID were provided in the Figure captions (Figures 3, 6, and 11 in the original manuscript). Patients ID have now been added as requested in the revision to the captions of Figures 12, 13, and 14.
Image quality. The Figures look really blurry in the PDF file for per-reviewing. I can assure that the images are much clearer in the Word.docxs version, which we received for the revision. In addition, we have submitted a Zip-File with the high-resolution images. I hope that these files will be used for the final preparation of manuscript by Editorial Office.
Visibility of tumors in Figure 6. The upper setting of this Figure was adjusted to SUV 3 to show clearly the superiority of the tumor uptake after injection of 80 μg DB8. However, the image with these settings does not support the statement “All primary lesions were visualized by SPECT/CT at all time points during the day of injection (2, 4, and 6 h after injection) of [99mTc]Tc-DB8”. Therefore, we prepared the image of the same patients with the upper setting of the scale adjusted to SUV 2.0 and placed it to Supplementary data. We added to the caption of Figure 6 the following text: “Note! The upper setting in this Figure was selected to show clearly the superiority of the tumor uptake after injection of 80 μg DB8. Supplementary Figure 3 shows tumors in the same image are clearly visualized after injection of [99mTc]Tc-DB8 with 40 and 120 μg DB8 with the upper setting of the scale adjusted to SUV 2. “
- What could be the reason for the significantly lower uptake at the 120 µg dose compared to 40 µg? Do the authors propose that further dose optimization in the range of 80 µg to 120 µg might be beneficial?"
Answer: We proposed the reason for the lower uptake in the Discussion section on the original manuscript “The further increase of the injection mass did not improve the tumor uptake or tumor-to-organ ratios, presumably due to the partial saturation of binding sites in tumors by the cold DB8. “ To make it more clear, we have re-written it as ”The further increase of the injection mass to 120 µg did not improve the tumor uptake or tumor-to-organ ratios, presumably due to the partial saturation of binding sites in the tumors by unlabeled DB8.“ In principle, any further optimization is welcome and desirable. However, practical considerations should be taken into account. For a translation into daily clinical practice, a kit formulation would be utterly desirable. A preparation of injected mass for each individual patient would be hardly possible in the clinical routine. Thus, we have to find the best peptide mass to lyophilize for a kit preparation in a vial. We have to take into account that the body weight of patients will vary. Thus, the exact finding of an optimal mass would not make much sense, but rather an optimal mass range seems to be more meaningful.
- Line 375-378; “There was no significant difference (p = 0.3, paired t-test) between tumor uptake at 2 h (SUVmax = 5.3 ± 1.4) and 4 h (SUVmax = 4.8 ± 1.9) after injection of [99mTc]Tc-DB8 with 80 μg DB8, but the uptake at 6 h after injection (SUV max = 4.0 ± 1.6) was significantly lower.” This interpretation does not corroborate with Fig 7.
Answer: Thank you for pointing out this unclear issue. Figure 7 is focused on the comparison of the tumor uptake after injection of different peptide mass. The p-values were calculated by ANOVA test, since three groups were analyzed. The cited text relates to comparison of tumor uptake at different time points after injection of 80 μg DB8. A paired t-test was used for companions, since the uptake in the same tumors was measured. To clarify this further, we paced the graphic output of the test in the Supplementary data and called it out in the main text, rephrasing it as “To select an optimal time point for imaging, both tumor uptake and imaging contrast was compared at different time points after injection of [99mTc]Tc-DB8 with 80 μg DB8. There was no significant difference (p = 0.3, paired t-test) between tumor uptake at 2 h (SUVmax = 5.3 ± 1.4) and 4 h (SUVmax = 4.8 ± 1.9) after injection of [99mTc]Tc-DB8 with 80 µg DB8, but the uptake at 6 h after injection (SUV max = 4.0 ± 1.6) was significantly lower (Supplementary Figure e).”
Round 2
Reviewer 1 Report
Comments and Suggestions for Authors
The authors have addressed all my comments very well.
Reviewer 2 Report
Comments and Suggestions for Authors
Following the modifications by the authors, particularly with respect to referencing at a number of places, and then minor modifications of the text for clarity, I believe the paper should now be accepted. My only minor concern is the number of patients, but given the complexity of the study, and the results from the patients that were statistically significant, this is only minor.
Reviewer 3 Report
Comments and Suggestions for Authors
This revised version of the manuscript addresses reviewers' concerns satisfactorily therefore the manuscript is recommended for acceptance.